# The New Digital Economy and Sustainability: Challenges and Opportunities

**Albérico Travassos Rosário** [1,*] and **Joana Carmo Dias** [2]

1   The Research Unit on Governance, Competitiveness and Public Policies (GOVCOPP), Universidade Europeia, 1200-649 Lisbon, Portugal
2   Centro de Investigação em Organizações, Mercados e Gestão Industrial (COMEGI), Universidade Lusíada, 1349-001 Lisbon, Portugal; joana.carmo.dias@universidadeeuropeia.pt
*   Correspondence: alberico@ua.pt

**Abstract:** This study aims to analyze the interconnection between the concepts of digital economy and sustainability. These concepts have become popular due to awareness of climate change and the increased development and adoption of technologies. Researchers, business leaders and policymakers are exploring the many ways digital technologies can be used to address sustainability issues. Using a systematic literature review with a bibliometric analysis, we examined a sample of 92 studies indexed in SCOPUS to identify research activity on this topic up until April 2023. We found that 2022 had the highest number of peer-reviewed articles, with 31 publications. During the research, we identified multiple opportunities for this interconnection, such as developing renewable energies and sustainable technological solutions, smart cities and sustainable urbanization, and sustainable consumption. These opportunities enabled by digital technologies allow companies to adopt sustainable business strategies and develop sustainable products. Despite these opportunities, the digital economy presents several challenges that can hinder efforts to achieve sustainability goals, such as increasing e-waste, high energy consumption and rising carbon emissions, the digital divide, job insecurity, growing monopolies, and data protection. These issues must be addressed to enable the optimal use of the opportunities presented in the digital economy to promote sustainability.

**Keywords:** new digital economy; sustainability; digital technologies; sustainable business strategies; sustainable products; bibliometric analysis



## 1. Introduction

In recent years, the rapid evolution of the digital economy has been facilitated by emerging disruptive technologies that have transformed business practices and consumption. These innovations changed technological, economic, and sociocultural phenomena, thus transforming conventional forms of commercial exchange. According to Pan et al. [1], the origin of the term "digital economy" dates back to the 1990s and refers to a form of economy originating from networked intelligence. However, the concept was officially recognized at the Hangzhou Summit of G20 leaders in 2016. Information technology is the core of the digital economy, while a modern network functions as the operator [1]. Thus, the digital economy is defined as the global network of economic activities, professional interactions, and commercial exchanges facilitated by information and communication technologies (ICT). It indicates the world's transition to the fourth industrial revolution. Technologies such as big data and analytics, artificial intelligence, machine learning, cloud computing, and virtual reality are changing the business landscape [2]. For example, these technologies require companies to modify their business models to accommodate the innovations. As a result, many markets have been disrupted, traditional businesses are under enormous pressure, and consumer behaviors and expectations have changed. Companies in the digital economy are restructuring their marketing strategies and means of

communication and consumer connection. Thus, digital technology creates opportunities for companies to improve efficiency and performance.

Conversations about the digital economy are often characterized by two main issues: the potential of ICTs and the impact on sustainability. Sustainable development is concerned with improving human well-being and is measured through crucial indicators such as satisfaction of needs, respect for human rights, security, social relations, and freedom of choice [3]. The technologies that characterize the digital economy contribute to sustainable development, providing innovations that help improve people's standard of living and providing advanced techniques to protect the planet, ensuring organizational profitability. In this case, ICT is integrated into the three pillars of sustainability—environmental, social, and economic [4]. However, the positive results of the digital revolution taking hold of global economies are associated with multiple sustainability concerns. While technology and the internet are a huge boost to sustainable development, they raise issues related to consumer protection, invasion of privacy, and cybersecurity issues [5]. As a result, most companies struggle to balance exploiting opportunities in the digital economy to support sustainable development while mitigating potential challenges that can undermine efforts. This systematic literature review with bibliometric analysis (SRLBA) provides insights into the opportunities and challenges of the digital economy in a sustainability context to support business professionals' efforts to integrate advanced technologies in a sustainable development context. This study is essential to understand how we can mitigate environmental impacts, optimize the use of resources, face climate change, promote social equity, and foster economic resilience. By incorporating sustainability principles into digital practices, we can harness the power of technology to build a more sustainable and inclusive future.

## 2. Materials and Methods

A systematic literature review with bibliometric analysis (SLRBA) was performed to identify relevant sources. A systematic literature review with a bibliometric analysis approach provides a robust and evidence-based understanding of a research topic. It allows researchers to identify research gaps, measure productivity and impact, visualize collaborations, make informed decisions, and stay abreast of current research trends. While the systematic literature review ensures a thorough and systematic assessment of the existing body of knowledge on a given topic with a well-defined search strategy as well as inclusion and exclusion criteria, a bibliometric analysis involves quantitative measures, such as citation counts, coauthorship networks, and publication trends, to assess research productivity and impact. Combining systematic review methodology and bibliometric analysis increases the credibility and reliability of the results, strengthening the evidence base for decision-making processes. This process ensures that only high-quality sources with defined methodologies are selected and synthesized in the review, resulting in representative and quality results. Given that this research aims to provide practical insights that can be adopted in business practice to aid decision making and strategy making, a detailed methodology such as the SLRBA is required.

To initiate the SLRBA, the academic database that helps identify relevant sources was identified, namely SCOPUS, for its recognition as one of the "largest databases of curated abstracts and citations" due to its global coverage of scientific journals, conferences, and books (p. 1) [6]. It also uses rigorous content screening and re-evaluation by an independent content screening and advisory board to ensure that only the highest-quality data are indexed.

The SLRBA involves screening and selecting information sources to ensure the validity and accuracy of data presented in a process consisting of three phases and six steps [7–9] (Table 1).

**Table 1.** Process of systematic literature review with bibliometric analysis.

| Phase | Step | Description |
| --- | --- | --- |
| Exploration | Step 1 | Formulating the research problem |
| | Step 2 | Searching for appropriate literature |
| | Step 3 | Critical appraisal of the selected studies |
| | Step 4 | Data synthesis from individual sources |
| Interpretation | Step 5 | Reporting findings and recommendations |
| Communication | Step 6 | Presentation of the SLRBA report |

The SCOPUS database was used to identify relevant sources for analysis. The search process started with the keyword "digital economy", resulting in 7870 results. Then, the exact keyword "sustainability" was added, which reduced the results to 92 documents (Table 2). These 92 documents are distributed in 62 articles, 13 conference papers, 11 book series, and 6 books.

**Table 2.** Screening methodology.

| Database SCOPUS | Screening | Publications |
| --- | --- | --- |
| Meta-search | Keyword: digital economy | 7870 |
| Inclusion Criteria | Keyword: digital economy<br>Exact keyword: sustainability | |
| Screening | Keyword: digital economy<br>Exact keyword: sustainability<br>Published until April 2023 | 92 |

Source: own elaboration.

## 3. Literature Analysis: Themes and Trends

Peer-reviewed documents from up until April 2023 were reviewed (Figure 1). The year 2022 was the year with the most peer-reviewed papers, with 31 publications. Publications were classified as follows: Swiss Sustainability (27); International Journal of Environmental Research and Public Health (5); Class Notes on Networks and Systems (4); Frontiers of Psychology (3); Economic Research Ekonomska Istrazivanja (2); Resource Policy (2); and the remaining publications, for which there was 1 document.

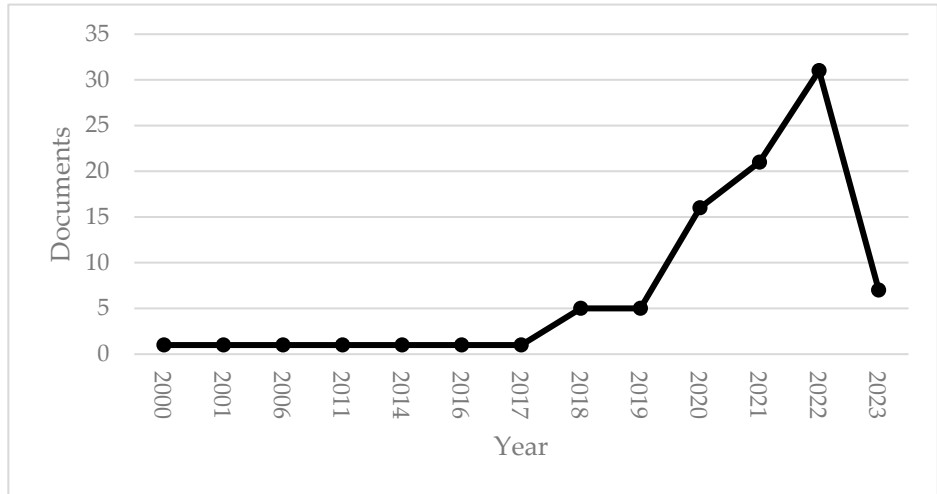

**Figure 1.** Documents by year. Source: own elaboration.

In Table 3, we analyze the Scimago Journal & Country Rank (SJR), the best quartile, and the h-index per publication. The best-ranked journal was Earth S Future with 2510 (SJR), Q1, and an h-index of 61. There were a total of 12 journals in Q1, 8 journals in Q2, 2 journals

in Q3, and 6 journals in Q4. Thus, the best quartile, Q1, represented 22%, Q2 represented 15%, Q3 represented 4%, and Q4 represented 3%. As evident from Table 1, the significant majority of articles on education entrepreneurship sustainability ranked in the Q1 best quartile index. The thematic areas covered by the 92 scientific and/or academic documents were: Social Sciences (48); Environmental Sciences (44); Computer Science (33); Engineering (31); Energy (30); Business, Management and Accounting (16); Economics, Econometrics and Finance (13); Medicine (7); Decision Sciences (4); Earth and Planetary Sciences (3); Psychology (3); Agricultural and Biological Sciences (2); Materials Science (1); Mathematics (1); and Pharmacology, Toxicology and Pharmaceuticals (1).

**Table 3.** Scimago Journal & Country Rank impact factor.

| Title | SJR | Best Quartile | H-Index |
|---|---|---|---|
| Earth S Future | 2.510 | Q1 | 61 |
| Technology In Society | 1.490 | Q1 | 69 |
| Resources Policy | 1.460 | Q1 | 80 |
| Annals Of Operations Research | 1.170 | Q1 | 111 |
| Frontiers In Public Health | 1.130 | Q1 | 80 |
| Frontiers In Environmental Science | 1.010 | Q1 | 61 |
| Frontiers In Psychology | 0.870 | Q1 | 133 |
| International Journal Of Environmental Research And Public Health | 0.810 | Q1 | 138 |
| Sustainability Switzerlan | 0.660 | Q1 | 109 |
| Economic Research Ekonomska Istrazivanja | 0.570 | Q2 | 35 |
| Advances In Civil Engineering Materials | 0.550 | Q2 | 15 |
| Journal Of Business Economics And Management | 0.500 | Q2 | 45 |
| Foresight | 0.480 | Q2 | 36 |
| Management And Marketing | 0.480 | Q2 | 19 |
| Interaction Design And Architecture S | 0.360 | Q1 | 16 |
| Regional Research Of Russia | 0.340 | Q1 | 15 |
| Izvestiya Rossiiskoi Akademii Nauk Seriya Geograficheskaya | 0.310 | Q1 | 12 |
| Journal Of Corporate Law Studies | 0.290 | Q2 | 11 |
| Investigaciones Regionales | 0.280 | Q2 | 20 |
| Frontiers In Artificial Intelligence | 0.250 | Q4 | 61 |
| Journal Of Environmental Protection And Ecology | 0.250 | Q3 | 25 |
| Journal Of Siberian Federal University Humanities And Social Sciences | 0.250 | Q2 | 10 |
| ACM International Conference Proceeding Series | 0.230 | -* | 128 |
| Ceur Workshop Proceedings | 0.230 | -* | 57 |
| Advances In Intelligent Systems And Computing | 0.220 | Q4 | 48 |
| Communications In Computer And Information Science | 0.210 | Q4 | 55 |
| Emerald Emerging Markets Case Studies | 0.190 | Q4 | 7 |
| Proceedings Of Institution Of Civil Engineers Energy | 0.190 | Q4 | 24 |
| Contaduria Y Administracion | 0.180 | Q3 | 16 |
| E3s Web Of Conferences | 0.180 | -* | 33 |
| Lecture Notes In Networks And Systems | 0.150 | Q4 | 22 |
| 2011 International Conference On E Business And E Government Icee2011 Proceedings | 0 | -* | 6 |
| Contributions To Conflict Management Peace Economics And Development | 0 | -* | 6 |
| Greener Management International | 0 | -* | 44 |
| Proceedings Of The 2nd World Conference On Smart Trends In Systems Security And Sustainability Worlds4 2018 | 0 | -* | 6 |
| Proceedings Of The 33rd International Business Information Management Association Conference Ibima 2019 Education Excellence And Innovation Management Through Vision 2020 | 0 | -* | 14 |
| Proceedings Of The Annual Hawaii International Conference On System Sciences | 0 | -* | 95 |
| Webist 2006 2nd International Conference On Web Information Systems And Technologies Proceedings | 0 | -* | 7 |
| 15th Annual IEEE International Systems Conference Syscon 2021 Proceedings | -* | -* | -* |
| 2022 IEEE Technology And Engineering Management Society Conference Asia Pacific Temscon Aspac 2022 | -* | -* | -* |
| Csr Sustainability Ethics And Governance | -* | -* | -* |
| Economics Law And Institutions In Asia Pacific | -* | -* | -* |
| Future Of Innovation And Technology In Education Policies And Practices For Teaching And Learning Excellence | -* | -* | -* |

**Table 3.** *Cont.*

| Title | SJR | Best Quartile | H-Index |
|---|---|---|---|
| Iccc 2022 IEEE 10th Jubilee International Conference On Computational Cybernetics And Cyber Medical Systems Proceedings | _* | _* | _* |
| Oxford Handbook Of Digital Technology And Society | _* | _* | _* |
| Oxford Handbook Of Industrial Hubs And Economic Development | _* | _* | _* |
| Oxford Handbook Of Luxury Business | _* | _* | _* |
| Preparing A Workforce For The New Blue Economy People Products And Policies | _* | _* | _* |
| Proceedings 2021 IEEE European Symposium On Security And Privacy Workshops Euro S And Pw 2021 | _* | _* | _* |
| Proceedings 2022 23rd International Arab Conference On Information Technology Acit 2022 | _* | _* | _* |
| Research For Development | _* | _* | _* |
| Scientific Horizons | _* | _* | _* |
| Social Business Models In The Digital Economy New Concepts And Contemporary Challenges | _* | _* | _* |
| Techno Review International Technology Science And Society Review Revista Internacional De Tecnologia Ciencia Y Sociedad | _* | _* | _* |
| World Sustainability Series | _* | _* | _* |

Note: * data not available. Source: own elaboration.

The most cited article was Governance Strategies for a Sustainable Digital World, published in *Sustainability* with 76 citations, an SJR of 0.660, the best quartile (Q1), and an h-index of 109. This article aims to analyze three governance strategies that countries can use with adaptive governance to respond to sustainability threats from digitalization.

The h-index was used to verify the productivity and impact of published works based on the most significant number of included articles with at least the same number of citations. Of the documents considered for the h-index, 12 were cited at least 12 times.

From Figure 2, we can analyze the evolution of document citations until April 2023. The number of citations shows a positive net growth, with an $R^2$ of 49% for 2022 (which has 289 citations), and a total of 591 citations overall. In Appendix A, Table A1, the citations of all scientific and/or academic documents until April 2023 are analyzed; 16 documents were not cited in this period, making a total of 517 citations.

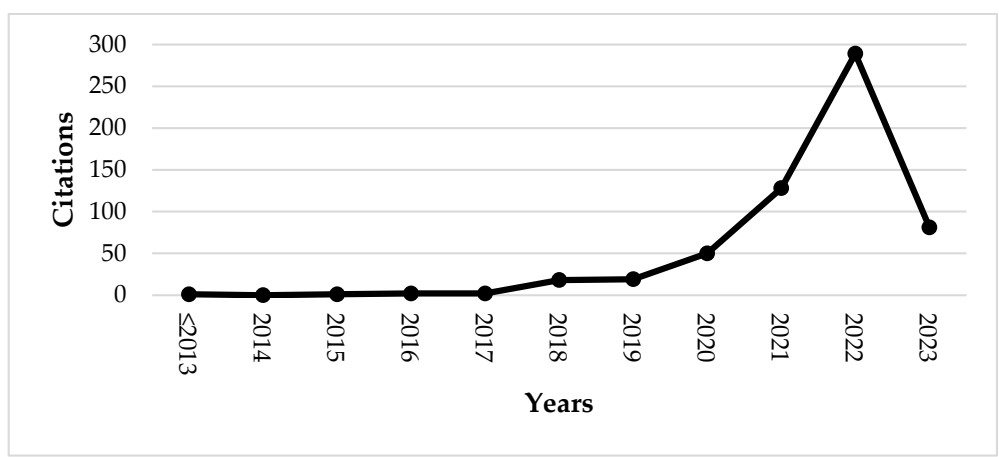

**Figure 2.** Evolution of citations between ≤2012 and December 2022. Source: own elaboration.

The study of bibliometric results, using the scientific software VOSviewer, aims to identify the main research keywords in studies related to the digital economy and sustainability. The node's size indicates the occurrence of the keyword, that is, the frequency with which the keyword appears. The connection between the nodes denotes the co-occurrence of the keywords; that is, they occur at the same time or together. Keyword co-occurrences are indicated by their thickness, that is, how often two or more keywords arise simulta-

neously. The keyword frequency increases with the node's size, the thickness of the links connecting the nodes, and the frequency of co-occurrences between the keywords. Each color represents a thematic cluster, where the nodes and links within the cluster can be used to explain the thematic coverage of the theme (represented by the nodes within the cluster) and the connections between the nodes that make up the theme (represented by the links within the cluster).

The research was based on the analyzed digital economy and sustainability articles. The associated keywords are presented in Figures 3 and 4, making clear the network of keywords that appear together/linked in each scientific article, thus allowing us to know the topics studied by the researchers and identify future research trends.

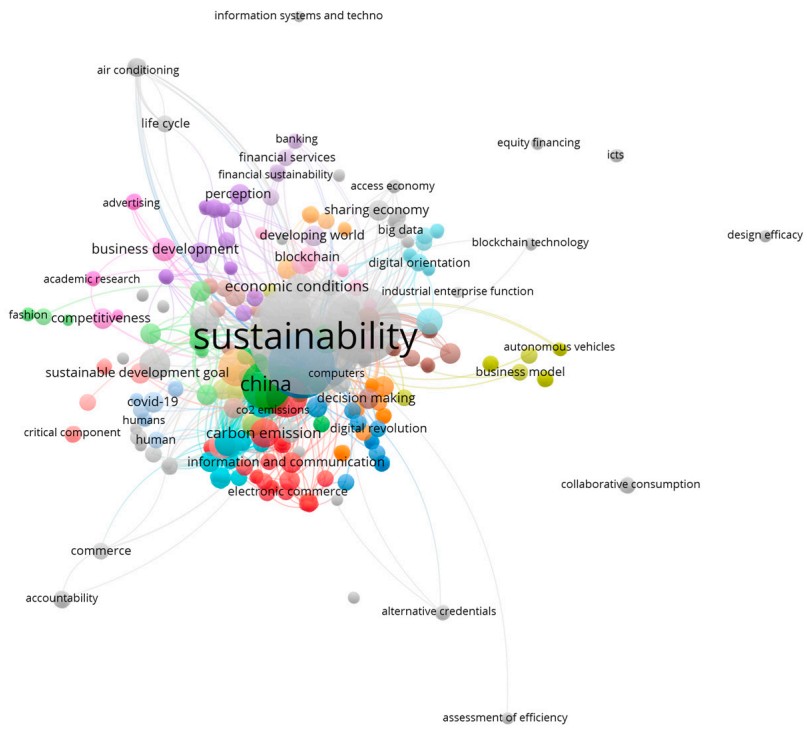

**Figure 3.** Network of all keywords. Source: own elaboration.

The biggest nodes in this mapping are sustainable development, marketing, and electronic commerce. The results of the keyword development map from the Vosviewer were divided into three clusters. Cluster 1 is grey with five keyword items, cluster 2 is green with five keyword items, and cluster 3 is blue with four keyword items, which can be seen in Figure 4 below. Cluster 1 is the largest cluster and refers to sustainable development. These articles mainly focus on digital marketing, business development, brands, B2B marketing, and analog-to-digital conversion. Cluster 2 refers to marketing and focuses on issues such as advanced technology, big data, artificial intelligence, strategic planning, and quantitative study. Cluster 3, electronic commerce, involves social media, social media platforms, social networking, and stakeholders. The three clusters are interconnected through the digital economy and sustainability themes. In Figure 5, we can analyze the profusion of co-citation.

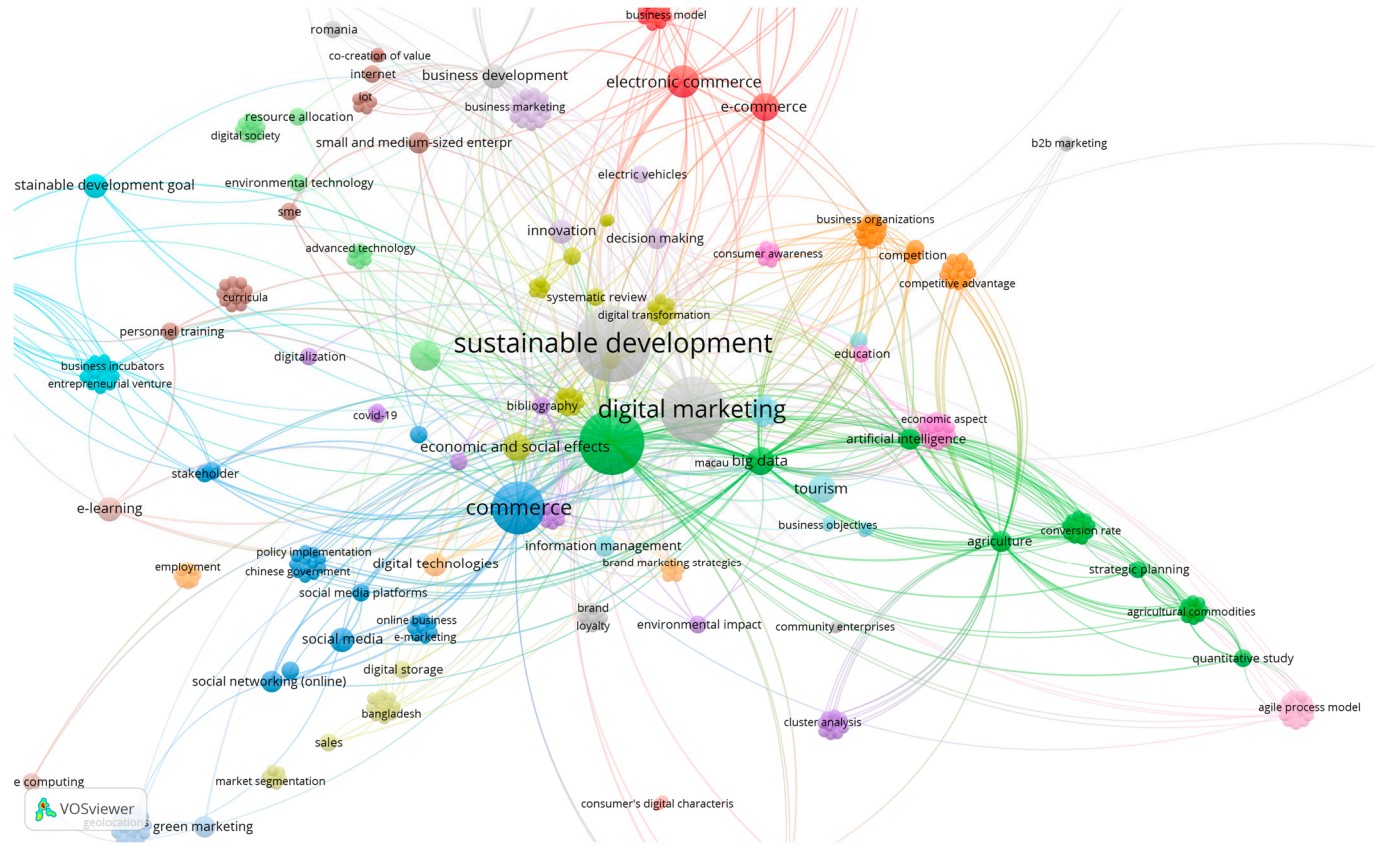

**Figure 4.** Network of linked keywords. Source: own elaboration.

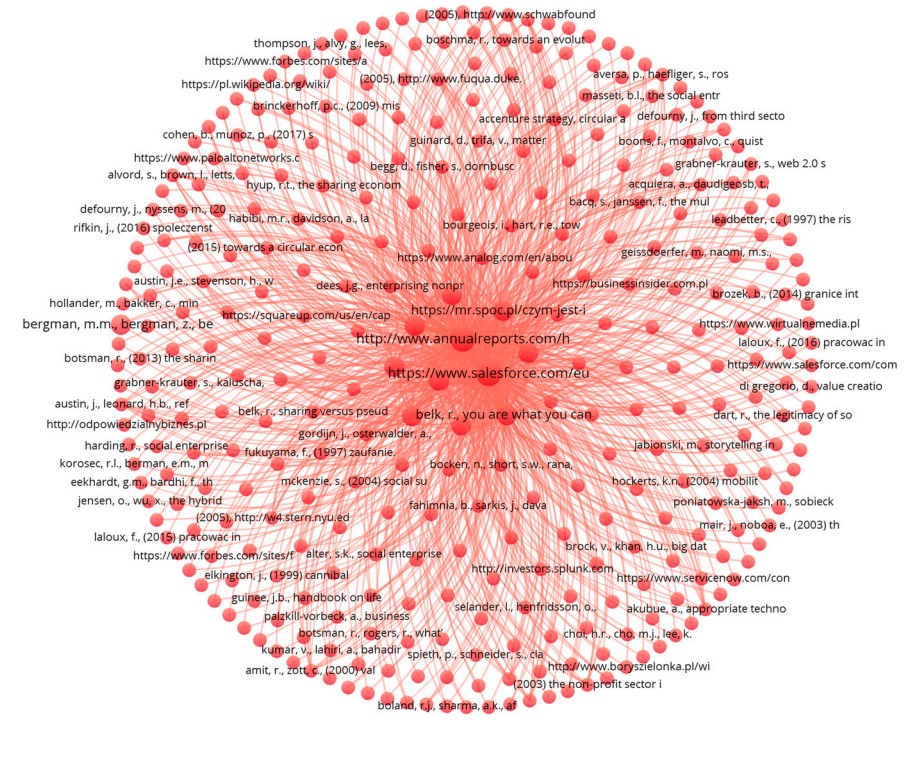

**Figure 5.** The profusion of co-citation. Source: own elaboration.

## 4. Theoretical Perspectives

The intersection of the digital economy and sustainability creates challenges and opportunities for entrepreneurs, policymakers, and society. For example, the digital economy promotes sustainable development through innovations such as green technologies and circular economy models [10]. It also offers opportunities that expand digital inclusion and empowerment. However, the rapid growth of the digital economy also presents several challenges, including high energy consumption, e-waste, and pollution. In recent years, there has also been an increase in cybersecurity issues, including consumer concerns about data privacy and security and cyberattacks [11]. Therefore, a comprehensive literature review is needed to assess the relationship between the digital economy and sustainability to create appropriate strategies that address the challenges, while enabling companies to seize the opportunities.

### 4.1. Conceptualization of the Digital Economy

The term "digital economy" refers to economic activities and transactions using digital technologies such as smartphones, the internet, computers, and tablets. Although the concept has become increasingly popular in the 2010s, its origin dates back to the creation of the internet and the emergence of personal computers in the 1980s [12]. The invention of the world wide web and of smartphones in the late 1980s and 1990s were important milestones in developing the digital economy. These technologies have continuously digitized and transformed the environment and business practices, causing significant transformations [13–15].

Several distinctive features characterize the digital economy and differentiate it from traditional economies. First, the digital economy has high connectivity and interdependence through networks and digital platforms [16]. For example, businesses and consumers can connect in real time, regardless of geographic distance or time zone differences, through digital technologies such as the internet and social media platforms. According to Rappitsch [17], 95% of the global population live in areas covered by a mobile cellular network. Therefore, the Internet has a high reach, which allows people from different regions to have access to content published online [18]. Second, the digital economy is characterized by the effective use of data [19]. Data technologies such as big data, analytics, and artificial intelligence fuel the growth of the digital economy. These innovations help generate and collect unprecedented data, increasing companies' access to insights into consumer behavior, market trends, and other business-critical information [20]. Third, the digital economy is characterized by disruptive technologies that facilitate rapid innovation and disruption. As a result, companies can produce and launch new products, services, and business models at a pace previously unimaginable.

The digital economy presents several benefits that can contribute to economic growth, innovation, and development. For example, it facilitates the creation and dissemination of knowledge [21]. As a result, it enabled new forms of collaboration and learning and new products and services, and expanded companies' access to new markets and opportunities. The digital economy is also associated with greater efficiency and productivity [22,23]. Companies are leveraging digital technologies to streamline their operations, improve the accuracy and speed of decision making, and automate repetitive tasks. For example, advanced communication channels facilitate quick communication and collaboration between business leaders, thus ensuring easy task execution and timely decision making [24]. Finally, the digital economy has created new jobs and employment opportunities, especially in technology development, digital marketing, and e-commerce.

### 4.2. Conceptualization of the Sustainability

The concept of sustainability has become increasingly popular in recent decades due to increased awareness and concerns about environmental issues such as climate change, biodiversity loss and social inequality. It was further popularized by the United Nations 2030 Agenda for Sustainable Development, which indicates that achieving sustainable

development requires consideration of three main dimensions: environmental, social, and economic [25]. Sustainable development is concerned with ensuring organizational financial performance while protecting people and the planet. In this case, sustainability can be defined as the capacity of a system to last a long time without degrading or depleting the resources on which it depends [25,26]. Thus, sustainability is concerned with meeting the needs of the current generation without compromising the ability of future generations to meet their own needs [27]. According to this concept, companies should adopt sustainable business models that exploit current resources using approaches that do not threaten the health and well-being of present and future generations [28]. As a result, global leaders and policymakers consider sustainability a fundamental principle underpinning a range of policies and governance structures at the local, national, and international levels.

Sustainability can be defined from three interrelated perspectives: environmental, social, and economic. Environmental sustainability concerns preserving and improving natural resources, including water, biodiversity, air, and soil [29]. Therefore, environmental sustainability includes strategies that help ensure that human activities do not exceed the Earth's capacity to support life and protect the resilience and adaptability of ecosystems. This dimension of sustainability focuses on maintaining an ecological balance in the planet's natural environment, conserving natural resources [30]. Social sustainability is about improving the well-being of people and communities, addressing issues such as equity, social justice, human rights, and cultural diversity [31]. As a result, it promotes equal access to basic needs such as food, water, shelter, and health care. In this case, the social dimension of sustainable development prioritizes building resilient infrastructure and promoting sustainable and inclusive economic growth that ensures equity.

Furthermore, social sustainability ensures that all people can fully participate in decisions that affect their lives [32]. Finally, economic sustainability is concerned with wealth creation and resource allocation, focusing on long-term viability and stability. This dimension promotes sustainable economic activities that help companies achieve goals without depleting natural resources or harming social welfare [33]. These dimensions of sustainability are crucial to promoting and achieving sustainable development and eradicating poverty. However, achieving sustainability requires collaboration across all sectors of society, including government, business, civil society, and individuals [34]. It requires shifting towards more sustainable practices and behaviors and adopting innovative technologies and governance models that support sustainability.

### 4.3. The Relationship between Digital Economy and Sustainability

The concepts of digital economy and sustainability are closely linked. The digital economy is characterized by technologies that can help reduce carbon emissions and promote circular economies, thus supporting sustainability. Esses et al. [35] explain that strategies and solutions based on digital technology increase environmental sustainability through areas such as pollution control and sustainable urban development, transport, and production systems. Advanced technologies in the digital economy empower business leaders and policymakers to assess and monitor business activities and regulations to promote sustainable development [36,37]. For example, companies can use these technologies to monitor their carbon footprint and employ measures to reduce emissions.

Similarly, policymakers can use data collected and analyzed using digital technologies to establish legal frameworks that encourage rapid transition to sustainable business practices [25]. These cases indicate that sustainability can be achieved by adopting digital technologies that improve the sustainable exploitation of resources and allow the generation of renewable energy. For example, energy companies can integrate advanced innovations to build smart grid technologies that can help optimize electricity distribution, promote renewable energy generation, and reduce waste [38]. These instances show that the digital economy and sustainability are increasingly intertwined.

There are many ways in which the digital economy can enable sustainability. For example, resource efficiency is an essential principle of the digital economy. Companies are

taking advantage of new digital technologies to deliver more value with less input, resulting in the efficient use of resources [39,40]. For example, innovative solutions in the digital economy can help reduce the use of natural resources and waste. Companies using internet of things (IoT) technology can access the real-time monitoring of energy usage [41]. Another way the digital economy supports sustainability is by promoting the circular economy. The circular economy model emphasizes the reuse and regeneration of materials and products to sustainably continue production [42,43]. It contributes to sustainability by closing and narrowing resource loops through durable product design, reuse, remanufacturing, repair, renovation, and recycling [44]. This means that the circular economy can reduce waste and pollution by always keeping products, components and materials at their highest value and usefulness. Digitizing the circular economy through innovative solutions helps to improve resource efficiency, product lifespan, customer relationships, and resilience [45]. For example, leveraging digital platforms such as social media and e-commerce sites can enable the sharing and exchange of products and resources. In this case, customers can sell or exchange a product they are not using, thus reducing waste, and promoting resource efficiency [46]. Likewise, companies can leverage these platforms to collect these products or materials that can be used to remanufacture other products or be refurbished or repaired for resale.

The digital economy creates new opportunities for economic growth while minimizing environmental impacts, thereby supporting sustainable development. For example, the digital economy is characterized by an increased development and adoption of renewable energy technologies, consequently minimizing dependence on fossil fuels, and reducing greenhouse gas emissions [47]. Likewise, digital technologies can enable the development of smart cities characterized by sustainable development practices, thus improving the quality of life of urban residents and communities. Finally, the digital economy promotes sustainable consumption [48]. Modern consumers have easy access to information about the environmental impacts of products and services [49]. For example, smart packaging uses QR codes that allow customers to scan and access more product information, including sourcing and production. This practice helps them make informed consumption or purchase decisions, especially for customers who value sustainability and ethical business practices [50]. Furthermore, digital technologies such as blockchain improve transparency and accountability in supply chains, promoting sustainable production and consumption patterns [51]. Therefore, digital technologies that facilitate the growth of the digital economy present several innovative solutions and technologies that can help achieve sustainability.

### 4.4. Opportunities in the Digital Economy for Sustainability

The digital economy creates new opportunities for innovation, collaboration and empowerment that drive sustainability. With the world facing severe environmental and social challenges such as climate change and inequality, it has become essential to harness the power of digital technologies to develop a more sustainable and equitable [52]. This section explores several opportunities identified in the research through which the digital economy promotes sustainability (Figure 6).

### 4.4.1. Renewable Energy and Sustainable Technology Solutions

Power generation and consumption are major issues at the heart of sustainability, as they are associated with adverse environmental issues such as climate change, air and water pollution, and solid waste disposal [53]. The digital economy addresses these problems by presenting innovative opportunities for renewable energies, contributing to sustainability [54]. For example, organizations use digital technologies for efficient and effective monitoring and control of renewable energy systems [55]. Furthermore, the digital economy is characterized by smart grids, which use digital technologies to manage energy distribution and strengthen dependence on renewable energy as a primary source of energy.

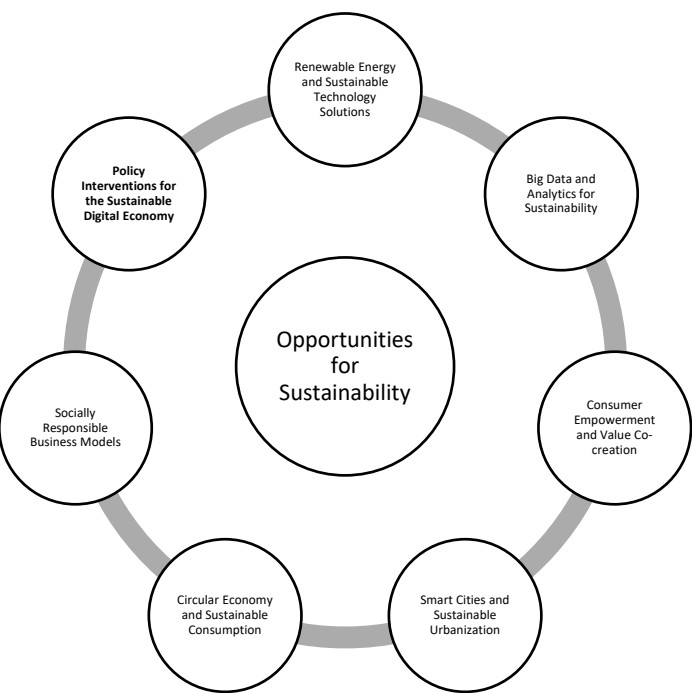

**Figure 6.** Opportunities in the digital economy for sustainability.

The digital economy promotes responsible production and consumption through technological solutions. An example of such a solution is 3D printing, a digital manufacturing technology, to reduce waste and increase efficiency by facilitating on-demand production instead of mass production [56,57]. Advanced digital technologies in today's digital economy allow companies to create sharing and renting platforms that promote sustainable consumption patterns [58]. These digital platforms enable the circular economy model, thus promoting sustainable development.

4.4.2. Big Data and Analytics for Sustainability

Big data and analytics help collect and analyze large amounts of data to assist in making informed decisions [59]. One of the main ways that big data and analytics in the digital economy contribute to sustainability is by providing advanced tools to monitor and track environmental impacts. Companies can access data on factors such as greenhouse gas emissions, energy consumption, and water use through advanced analytical tools [60,61]. Data can be analyzed to gather information that allows an organization to determine areas to reduce environmental impact [62]. This practice can result in greater resource efficiency, lower energy use, and the greater use of renewable energy. Additionally, companies can take advantage of opportunities created through big data and analytics to optimize supply chains for sustainability [63].

4.4.3. Consumer Empowerment and Value Cocreation

Several authors [64,65] explain that innovation and competition in the digital economy triggers customer well-being through various aspects, including ever-changing customer demands and the rise of consumer protection laws. As mobile technologies advance, consumers' online skills, awareness, and engagement increase [66]. As a result, they can better evaluate information correctly before making optimal purchasing or consumption decisions, including choosing environmentally friendly products and services. An empowered consumer can drive innovation and increase competition and productivity, resulting in sustainable development.

In addition, digital technologies offer value cocreation opportunities, where companies collaborate with customers to develop products and services. This approach helps

ensure that final products meet customer needs and promote sustainability [67]. Involving customers throughout the value creation process provides brands with access to different perspectives, increases idea and opportunity generation, and enhances collaboration, building loyalty and trust [68,69].

### 4.4.4. Smart Cities and Sustainable Urbanization

Urbanization and industrialization have increased the number of people living in cities. This has increased carbon emissions, as evidenced in research by Bull and Azennoud [70], who argue that the population living in cities contributes two-thirds of global carbon dioxide emissions. The digital economy presents technologies for building smart cities and sustainable urbanization, thus improving the quality of life of its residents and reducing the environmental impact [71,72]. For example, the city can install smart sensors to monitor air quality, traffic, and energy consumption. The collected data can be analyzed and interpreted to assist in making informed decisions about resource management.

Furthermore, sustainable urbanization can leverage digital platforms to influence residents' sustainable living practices. For example, online platforms can connect citizens with information about sustainable living practices and resources. Bull and Azennoud [73] report that 30% of a building's energy is wasted through the behaviors of its residents. City management can use the internet and social media platform to raise awareness and encourage behavioral changes to improve energy efficiency [74]. Furthermore, these communication platforms can raise awareness of the need for sustainable resources such as public transport, green spaces, and recycling facilities [75].

### 4.4.5. Circular Economy and Sustainable Consumption

The dominant business model follows a throwaway strategy, which continues to cause significant environmental problems such as improper waste management. This challenge is evidenced in the commentary by Stahel [76], which indicates that one-third of global plastic waste is not collected or managed. The circular economy and sustainable consumption address this issue, encouraging a new model that replaces disposal with sustainable practices such as reuse, recycling, repair, and remanufacturing [77]. According to Stahel [78], the adoption of the circular economy model can reduce each country's greenhouse gas emissions by 70% and increase job opportunities by 4%. Likewise, sustainable consumption can have comparable impacts on sustainability, as it encourages consumers to choose products and services that are environmentally friendly.

Furthermore, advanced innovations such as the Internet of Things (IoT) can improve resource efficiency [79]. For example, they can monitor resource usage in real time, thereby identifying inefficiencies and resource recovery opportunities.

### 4.4.6. Socially Responsible Business Models

In this era of global access to knowledge and information, modern companies understand the importance of incorporating social aspects into their business models. For example, the long-term success and sustainability of modern companies depend on the social acceptance of proposed solutions, including products and services [80]. As a result, socially responsible business models have become increasingly popular as they encourage companies to incorporate social and environmental considerations into decision making and business operations [81]. Consumers demand environmentally friendly products and hold companies accountable for the impacts of their business activities on the environment and society. Therefore, the adoption of socially responsible business models has become a way for companies to develop and promote "socially acceptable" products, since they incorporate consumers' sustainability concerns [82]. As a result, these models create value for all stakeholders, including communities, customers, employees, and the environment.

In addition, the digital economy promotes socially responsible business models through digital platforms that increase transparency and accountability in business operations. For example, it provides digital technologies to help companies track and report

social and environmental performance, identifying opportunities to improve their sustainability practices [83].

### 4.4.7. Policy Interventions for the Sustainable Digital Economy

As the digital economy rapidly evolves, policymakers must develop regulatory frameworks that enable economic growth and innovation while ensuring social and environmental well-being. For example, local and international agencies have developed regulations and standards that promote sustainable digital economy practices, such as policies to reduce the carbon footprint [84,85]. In addition, some countries have standards to promote the use of sustainable materials in producing digital devices and equipment. Governments are also investing in research and development of sustainable digital technologies and providing incentives for companies that adopt these innovations [86].

### 4.5. Challenges in the Digital Economy in Sustainability

Despite its benefits and opportunities, the digital economy presents significant challenges. For example, as digital technologies and platforms dominate various sectors, there are growing concerns about their negative impacts on the environment, society, and economy [87]. For example, some people are concerned about energy demand and consumption when running facilities like data centers, which are the backbone of the data-driven digital economy. Thus, it is necessary to assess the potential challenges that characterize the digital economy in sustainability to develop a more sustainable and responsible approach and promote innovation, growth, and sustainability (Figure 7).

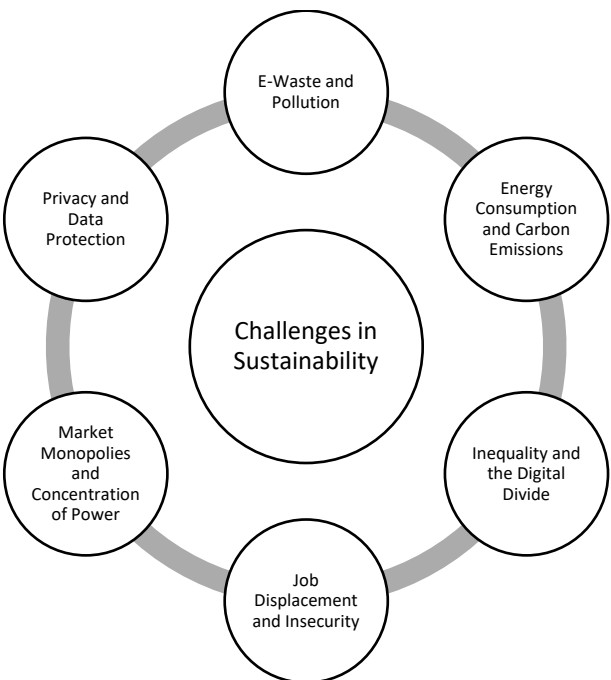

**Figure 7.** Challenges in the Digital Economy in Sustainability.

### 4.5.1. E-Waste and Pollution

The digital economy is characterized by rapid innovation and the emergence of new technologies that generate more e-waste, creating significant environmental and health risks. The lack of proper e-waste management systems compounds the problem, as most nations lack adequate infrastructure and resources to dispose of e-waste [88]. Unsustainable solutions, such as dumping e-waste in landfills or illegally exporting it to developing countries, are commonplace. Notably, most electronic waste is toxic and non-biodegradable [89]. When methods such as open-air burning and acid baths extract valuable materials from electronic components, toxic components are released into the

environment, causing air, water, and soil pollution [90]. These components include mercury, polybrominated flame retardants, lead, barium, cadmium, and lithium, which can have profound health implications in surrounding communities.

### 4.5.2. Energy Consumption and Carbon Emissions

Electronic devices that support or increase the digital economy are associated with higher energy consumption or contribute even more to greenhouse gas emissions. For example, data centers typically consume a lot of electricity to power their servers, cool their facilities and provide backup power [91]. Furthermore, as the demand for high-performance devices and data-intensive applications continues to grow, technology manufacturers are producing and using more electronic devices [54]. These processes require significant energy inputs to facilitate activities such as extracting and processing raw materials and manufacturing and transporting devices [92]. Furthermore, high energy consumption and carbon emissions from the increased use of cloud-based services, media streaming, and other data-intensive applications have become a crucial challenge in the digital economy [20]. Therefore, more research and development are needed to balance the opportunities created by digital technologies and their consequent environmental impacts.

### 4.5.3. Inequality and the Digital Divide

While the digital economy aims to increase everyone's access to information, knowledge, and opportunity, it is characterized by high inequality and the digital divide. For example, Rappitsch [17] indicates that while online opportunities are accessible to at least 80% of the population in developed countries, only 15% of people in less developed countries have access to the Internet. Furthermore, 30 out of every 100 people have fixed broadband subscriptions in developed countries, compared to 0.7 in Africa. These statistics indicate that those with limited access to digital technologies are being left behind as the digital economy grows, resulting in disparities in economic, social, and educational opportunities. The lack of infrastructure and technologies limits these people's ability to participate in the digital economy and access various options, including education, employment, and health [93]. Furthermore, these regions and communities tend to have low literacy and digital skills, especially among older adults and low-income communities [94].

This challenge undermines the ability of the digital economy to achieve the sustainable development goals of increasing equity and equality to eradicate poverty and improve the well-being of the global population.

### 4.5.4. Job Displacement and Insecurity

Digital technologies and automation are advancing rapidly, increasing the risk of job losses, and changing the job market. For example, manufacturing and agriculture industries that rely heavily on manual labor are replacing workers with robots and automation technologies [95]. These changes significantly impact individuals, communities and the environment and exacerbate existing inequalities. Furthermore, there are growing concerns about the potential for a skills gap, as employees may lack the necessary skills to adapt to new technologies and the changing job market [96]. With innovations appearing consistently, it has become a challenge for employees to maintain the desired skills to explore the opportunities presented [97]. This situation has led to a mismatch between workers' skills and those in demand, leading to unemployment and underemployment [98,99]. The loss of jobs has a negative impact on the quality and standards of living, affecting the well-being of individuals.

### 4.5.5. Market Monopolies and Concentration of Power

As companies expand into international markets, concerns about the impact of market monopolies and the concentration of power are growing. Most large corporations have access to digital technologies and platforms and highly skilled talent, which allows them to continue to grow and dominate various sectors [100,101]. This dominance is often as-

sociated with the corporation's ability to limit competition, stifle innovation, and reduce consumer choice, negatively impacting the economy and social and environmental sustainability [102,103]. For example, monopolies can limit the diversity of products and services available to consumers and reduce incentives for sustainable practices [104]. Additionally, they may misuse or abuse their power through unethical business practices, manipulative algorithms, or data breaches. Low competition due to the concentration of power allows them to engage in unethical and unsustainable business practices without fear of retaliation, as consumers have limited access to alternative products or services. This challenge affects society and the environment, as evidenced by issues such as the exploitation of workers or the misuse of personal data.

### 4.5.6. Privacy and Data Protection

Privacy and data protection are essential issues in the digital economy that can undermine efforts to achieve sustainability. Most companies use digital information and communication technologies to collect, process and share large amounts of personal data, leading to privacy and data protection concerns [105]. For example, there have been numerous cases of unauthorized access to consumer personal information, data breach or sale of personal data to third parties [106]. These issues affect people's trust in digital technologies, ultimately impacting the adoption of sustainable practices.

### 5. Conclusions

The digital economy and sustainability are two concepts that can work together and result in beneficial gains for companies. Technologies that are characteristic of the digital economy, such as AI and IOT, present several innovative solutions that can help achieve sustainability. For instance, technological advances in digital technologies that drive the growth of renewable energy such as solar and wind energy; big data and analytics that help collect and analyze large amounts of data to assist in making informed decisions; the empowerment of consumers who now have access to information on commercial practices; and technologies for building smart cities and sustainable urbanization. These opportunities allow business leaders to embrace sustainable business strategies and develop sustainable products, thus contributing to sustainable development. In turn, policymakers, in the face of the growth of the digital economy, must develop regulatory frameworks that enable economic growth and innovation. However, the digital economy presents several challenges that can hinder efforts to achieve sustainability goals, such as increasing e-waste and pollution, high energy consumption, and rising carbon emissions. Other challenges include inequality and the digital divide, job insecurity, growing monopolies, the concentration of power among a few large corporations, and data protection and privacy concerns [103]. Therefore, these issues must be addressed to enable the optimal use of the opportunities presented in the digital economy to promote sustainability.

The digital economy has played, and will continue to play, a crucial role in sustainability, supporting the more efficient use of resources, optimizing energy use in buildings and factories, reducing food waste in supply chains, and improving the efficiency of transport networks. In addition, they can enable new business models and products that support sustainability, platforms that facilitate peer-to-peer sharing of goods and services, can promote the conservation of resources and products that use recycled materials, can reduce waste, and contribute to a circular economy that allows for greater transparency and accountability in sustainability efforts. Thus, we can conclude that the digital economy is increasingly recognized as having an essential role in promoting sustainability.

We can point out some limitations to the present study, namely the selection and use of databases and the chosen keywords. Although Scopus is the most extensive database, there are publications indexed in other databases, such as EBSCO and ISI Web of Science, that can be extremely important and allow a broader view of the results. As for the keywords used in the research, we admit that using only two keywords—digital economy

and sustainability—can reduce the search, so for future research, it would be essential to consider other related keywords.

For future research, we point out some possible lines of research that will potentiate the intersection between digital economy and sustainability: (i) digital solutions to sustainability challenges (exploring ways to reduce energy consumption in data centers); (ii) the impact of the digital economy on sustainable development (such as poverty reduction, access to education and health, and environmental protection); (iii) digital sustainability metrics and standards that can help companies and policymakers make more informed decisions; and (iv) the role of regulation in promoting digital sustainability (exploring the effectiveness of measures and identifying new ways in which regulation can support sustainable digital practices). In addition, other authors [104–106] suggest new lines of research such as the development of different frameworks for integrating operational excellence methods with Industry 4.0 technologies and integrating Green Lean Six Sigma (GLS) in Industry 4.0 to mitigate carbon footprints and produce high specification products.

**Author Contributions:** Conceptualization, A.T.R. and J.C.D.; methodology, A.T.R. and J.C.D.; software, A.T.R. and J.C.D.; validation, A.T.R. and J.C.D.; formal analysis, A.T.R. and J.C.D.; investigation, A.T.R. and J.C.D.; resources, A.T.R. and J.C.D.; data curation, A.T.R. and J.C.D.; writing—original draft preparation, A.T.R. and J.C.D.; writing—review and editing, A.T.R. and J.C.D.; visualization, A.T.R. and J.C.D.; supervision, A.T.R. and J.C.D.; project administration, A.T.R. and J.C.D.; funding acquisition, A.T.R. and J.C.D. All authors have read and agreed to the published version of the manuscript.

**Funding:** "This work was financially supported by the Research Unit on Governance, Competitiveness and Public Policies (UIDB/04058/2020) + (UIDP/04058/2020), funded by national funds through FCT—Fundação para a Ciência e a Tecnologia.", and the second receives financial support from Fundação para a Ciência e Tecnologia (through project UIDB/04005/2020).

**Institutional Review Board Statement:** Not applicable.

**Informed Consent Statement:** Not applicable.

**Data Availability Statement:** Not applicable.

**Acknowledgments:** We would like to express our gratitude to the editor and the referees. They offered valuable suggestions or improvements. The authors were supported by the GOVCOPP Research Center of the University of Aveiro and COMEGI—Centro de Investigação em Organizações, Mercados e Gestão Industrial da Universidade Lusíada.

**Conflicts of Interest:** The funders had no role in the design of the study; in the collection, analyses, or interpretation of data; in the writing of the manuscript; or in the decision to publish the results.

## Appendix A

**Table A1.** Overview of document citations period 2013 to 2023.

| Documents | | 2013 | 2014 | 2015 | 2016 | 2017 | 2018 | 2019 | 2020 | 2021 | 2022 | 2023 | Total |
|---|---|---|---|---|---|---|---|---|---|---|---|---|---|
| Research on Theoretical Mechanism and Promotion Path of Digi. | 2023 | - | - | - | - | - | - | - | - | - | - | 1 | 1 |
| How to improve environment, resources and economic efficienc... | 2023 | - | - | - | - | - | - | - | - | - | - | 2 | 2 |
| Innovation and Optimization Logic of Grassroots Digital Gove... | 2022 | - | - | - | - | - | - | - | - | - | - | 1 | 1 |
| Complex Network-Based Evolutionary Game for Knowledge Transf... | 2022 | - | - | - | - | - | - | - | - | - | - | 1 | 1 |
| Going Abroad and Going Green: The Effects of Top Management... | 2022 | - | - | - | - | - | - | - | - | - | - | 1 | 1 |
| Bibliometric Analysis of the Research on the Impact of Envir... | 2022 | - | - | - | - | - | - | - | - | - | 1 | 1 | 2 |

**Table A1.** *Cont.*

| Documents | | 2013 | 2014 | 2015 | 2016 | 2017 | 2018 | 2019 | 2020 | 2021 | 2022 | 2023 | Total |
|---|---|---|---|---|---|---|---|---|---|---|---|---|---|
| Digital Economy and Environmental Sustainability: Do Informa... | 2022 | - | - | - | - | - | - | - | - | - | 2 | 2 | 4 |
| THE IMPACT OF INNOVATION FRAMEWORK CONDITIONS ON CORPORATE... | 2022 | - | - | - | - | - | - | - | - | - | 4 | - | 4 |
| Editorial: Sustainable digital economy, entrepreneurship, an... | 2022 | - | - | - | - | - | - | - | - | - | - | 1 | 1 |
| EVOLUTION OF PROJECT MANAGEMENT IN THE DIGITAL ECONOMY | 2022 | - | - | - | - | - | - | - | - | - | 1 | - | 1 |
| Impact of Digital Finance on Regional Carbon Emissions: An E... | 2022 | - | - | - | - | - | - | - | - | - | 5 | 2 | 7 |
| Approaches Toward Building the Digital Enterprise and Sustai... | 2022 | - | - | - | - | - | - | - | - | - | 1 | - | 1 |
| How Does New Infrastructure Investment Affect Economic Growt... | 2022 | - | - | - | - | - | - | - | - | - | 1 | - | 1 |
| Transition to Digital Entrepreneurship with a Quest of Susta... | 2022 | - | - | - | - | - | - | - | - | - | 4 | | 4 |
| The Impact of COVID-19 Epidemic on the Development of the Di... | 2022 | - | - | - | - | - | - | - | - | - | 6 | 1 | 7 |
| Key Drivers of Urban Digital Economy Sustainable Developmen... | 2022 | - | - | - | - | - | - | - | - | - | 1 | - | 1 |
| Current Status and Challenges of Green Digital Finance in Ko... | 2022 | - | - | - | - | - | - | - | - | - | - | 1 | 1 |
| A decision framework for incorporating the coordination and... | 2022 | - | - | - | - | - | - | - | - | - | 1 | - | 1 |
| CARBON EMISSIONS AND THE DEVELOPMENT OF DIGITAL ECONOMY: A P... | 2022 | - | - | - | - | - | - | - | - | - | 3 | - | 3 |
| Sustainable digital economy and trade adjusted carbon emissi... | 2022 | - | - | - | - | - | - | - | - | - | 7 | 2 | 9 |
| Linkage Between Inclusive Digital Finance and High-Tech Ente... | 2021 | - | - | - | - | - | - | - | - | - | 8 | 3 | 11 |
| Digital transformation and sustainable oriented innovation:... | 2021 | - | - | - | - | - | - | - | - | 1 | 8 | 2 | 11 |
| The convergence model of education for sustainability in the... | 2021 | - | - | - | - | - | - | - | - | - | 6 | - | 6 |
| Funding sustainable online news: Sources of revenue in digit... | 2021 | - | - | - | - | - | - | - | - | - | 2 | 1 | 3 |
| Contextuality and Intersectionality of E-Consent: A Human-ce... | 2021 | - | - | - | - | - | - | - | - | - | 2 | - | 2 |
| Digital economic development and its impact on economic grow... | 2021 | - | - | - | - | - | - | - | - | 1 | 19 | 5 | 25 |
| The COVID-19 Pandemic Not Only Poses Challenges, but Also Op... | 2021 | - | - | - | - | - | - | - | - | 2 | 18 | 6 | 26 |
| Digital financial inclusion sustainability in Jordanian cont... | 2021 | - | - | - | - | - | - | - | - | 6 | 11 | 3 | 20 |
| Sustainability and digital transformation in the visegrad gr... | 2021 | - | - | - | - | - | - | - | - | 4 | 17 | 4 | 25 |
| Towards sustainable digital innovation of SMEs from the deve... | 2021 | - | - | - | - | - | - | - | - | 4 | 17 | 7 | 28 |
| Women entrepreneurship and sustainable business development:... | 2021 | - | - | - | - | - | - | - | - | 2 | 3 | 4 | 9 |
| Amazon's initiative transforming a non-contact society—Dig... | 2021 | - | - | - | - | - | - | - | - | - | 3 | 1 | 4 |

**Table A1.** *Cont.*

| Documents | | 2013 | 2014 | 2015 | 2016 | 2017 | 2018 | 2019 | 2020 | 2021 | 2022 | 2023 | Total |
|---|---|---|---|---|---|---|---|---|---|---|---|---|---|
| Dependability and Sustainability Evaluation of Data Center E... | 2021 | - | - | - | - | - | - | - | - | 1 | - | - | 1 |
| Human capital in digital economy: An empirical analysis of c... | 2021 | - | - | - | - | - | - | - | - | 6 | 16 | 9 | 31 |
| Exploring the sustainability of the intermediary role in blo... | 2021 | - | - | - | - | - | - | - | - | 2 | 9 | 1 | 12 |
| Australia's Blue Economy Cooperative Research Centre | 2021 | - | - | - | - | - | - | - | - | - | 1 | - | 1 |
| Innovative Development of the Digital Economy: A View of Sus... | 2021 | - | - | - | - | - | - | - | - | - | 3 | - | 3 |
| Objective sustainability assessment in the digital economy:... | 2021 | - | - | - | - | - | - | - | - | 2 | 3 | 1 | 6 |
| Specificity of sustainability assessment for industrial ente... | 2021 | - | - | - | - | - | - | - | - | 3 | 1 | - | 4 |
| Making Cyberspace towards Sustainability A Scientometric Rev... | 2020 | - | - | - | - | - | - | - | - | 1 | 1 | - | 2 |
| Green and Digital Economy for Sustainable Development of Urb... | 2020 | - | - | - | - | - | - | - | - | 2 | 4 | 3 | 9 |
| Emerging trends and drivers for knowledge-intensive economy | 2020 | - | - | - | - | - | - | - | - | - | 3 | - | 3 |
| Robust proof of stake: A new consensus protocol for sustaina... | 2020 | - | - | - | - | - | - | - | 3 | 13 | 14 | 4 | 34 |
| Sustainable business model based on digital twin platform ne... | 2020 | - | - | - | - | - | - | - | 2 | 10 | 17 | 4 | 33 |
| Companies and UN 2030 Sustainable Development Goal 9 Industr... | 2020 | - | - | - | - | - | - | - | 1 | 5 | 6 | - | 12 |
| Stability and Sustainability of Cryptotokens in the Digital... | 2020 | - | - | - | - | - | - | - | - | 2 | - | 1 | 3 |
| Green and digital economy as a means for sustainable develop... | 2020 | - | - | - | - | - | - | - | - | - | 1 | - | 1 |
| Methodological Approach to the Classification of Digital Eco... | 2020 | - | - | - | - | - | - | - | - | - | 3 | 1 | 4 |
| Digital future: Economic growth, social adaptation, and tech... | 2020 | - | - | - | - | - | - | - | - | 3 | - | 1 | 4 |
| India Towards Digital Revolution (Security and Sustainabilit... | 2019 | - | - | - | - | - | - | - | - | - | 1 | - | 1 |
| Social business models in the digital economy: New concepts... | 2019 | - | - | - | - | - | - | 1 | 2 | 5 | 3 | - | 11 |
| The problem of accounting non-economic characteristics when... | 2019 | - | - | - | - | - | - | - | 1 | - | - | - | 1 |
| Analysis of the maturity of sustainable project management i... | 2019 | - | - | - | - | - | 1 | - | 1 | 1 | - | - | 3 |
| Thai MOOC sustainability: Alternative credentials for digita... | 2019 | - | - | - | - | - | - | - | 1 | 1 | - | - | 2 |
| Material selection: Balancing sustainability and resilience | 2018 | - | - | - | - | - | - | 1 | - | - | - | - | 1 |
| Governance strategies for a sustainable digital world | 2018 | - | - | - | - | 7 | 12 | 18 | 13 | 25 | 1 | - | 76 |
| Agile Digital Skills Examination for the Digital Economy: Kn... | 2018 | - | - | - | - | - | - | - | 2 | - | - | - | 2 |
| The challenge of long-term tourism competitiveness in the a... | 2018 | - | - | - | - | - | - | 4 | 5 | 6 | 1 | - | 16 |
| Digitalisation and the UN Sustainable Development Geais: Wha... | 2018 | - | - | - | - | - | - | 5 | 9 | 10 | 1 | - | 25 |

**Table A1.** *Cont.*

| Documents | | 2013 | 2014 | 2015 | 2016 | 2017 | 2018 | 2019 | 2020 | 2021 | 2022 | 2023 | Total |
|---|---|---|---|---|---|---|---|---|---|---|---|---|---|
| Consumer empowerment in the digital economy: Availing sustai... | 2017 | - | - | - | - | 7 | 4 | 5 | 13 | 8 | 1 | - | 38 |
| Smart citizens for smart cities: Participating in the future | 2016 | - | - | 1 | 2 | 2 | 1 | 7 | 7 | 2 | - | - | 22 |
| Modularity and network integration: Emergent business models... | 2014 | - | 1 | - | - | 1 | 1 | - | 1 | - | - | - | 4 |
| A conceptual model for sustaining competitive advantage in... | 2006 | 1 | - | - | - | - | - | - | - | - | - | - | 1 |
| Balance of nature? Sustainable societies in the digital econ... | 2001 | - | - | 1 | - | - | - | - | 1 | - | - | - | 2 |
| | Total | 1 | 1 | 2 | 2 | 17 | 19 | 41 | 62 | 128 | 241 | 77 | 591 |

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
