# Peer review of "The New Digital Economy and Sustainability: Challenges and Opportunities"

_sustainability, doi:10.3390/su151410902_

Round 1

Reviewer 1 Report

The article is good but needs substantial improvement. Please find comments.

1.      Provide motivation for the study in the introduction section.

2.       There are a few typos and grammatical errors fix the same. The literature review needs to be updated in terms of sustainability and digital aspects. What are different dimensions etc? Here are a few suggestions in this regard. “Green Lean Six Sigma for sustainability improvement: a systematic review and future research agenda” “" Integration of Lean Manufacturing and Industry 4.0:” and Industrial revolution and environmental sustainability: an analytical interpretation of research constituents in Industry 4.0. "Integrating Green Lean Six Sigma and industry 4.0: a conceptual framework.

3.      Methodology section needs to be updated and needs a clear representation of the idea. Why authors have selected the present method and why not others?

4.      Results must be clear and discussed in detail related to the implications matters. How your results are comparable to the previous studies of the same nature must be presented.

5.      Implications need to be induced in terms of policymakers, practitioners and researchers

6.      Conclusion section needs to be revised in terms of the after-effects of the study.

fine

Author Response

Dear,

Reviewer

The article is good but needs substantial improvement. Please find comments.

  1. Provide motivation for the study in the introduction section.

Thank you for your feedback on the introduction section of the paper. We appreciate your perspective and agree that providing additional details on the motivation would be beneficial.

  1. There are a few typos and grammatical errors fix the same.

We appreciate the reviewer's comment and opportunity for improvement. We read the entire article to correct all typos and grammatical errors.

  1. The literature review needs to be updated in terms of sustainability and digital aspects. What are different dimensions etc? Here area few suggestions in this regard. “Green Lean Six Sigma for sustainability improvement: a systematic review and future research agenda” “" Integration of Lean Manufacturing and Industry 4.0:” and Industrial revolution and environmental sustainability: an analytical interpretation of research constituents in Industry 4.0. "Integrating Green Lean Six Sigma and industry 4.0: a conceptual framework.

Thank you for your kind comments on our manuscript. We appreciate your constructive comments to guide our revision work. Based on your suggestions, we have

focused on revising the Introduction section to more clearly clarify the research subject and its contribution to the literature.

  1. Methodology section needs to be updated and needs a clear representation of the idea. Why authors have selected the present method and why not others?

Thank you for your feedback on the methodology section of the paper. We appreciate your perspective and agree that providing additional details on the method would be beneficial.

  1. Results must be clear and discussed in detail related to the implications matters. How your results are comparable to the previous studies of the same nature must be presented.

Thank you for your careful Reading. Although we understand the comment, it is not practicable to make this correction due to the study methodology used here. A systematic literature review aims to provide a comprehensive and unbiased summary of existing research on a specific topic or research question. Thus, the results obtained from a systematic review can include, for instance, a synthesis of findings, recommendations and implications, making it impracticable to compare our results with the results of previous studies.

  1. Implications need to be induced in terms of policymakers, practitioners and researchers

This is certainly an important concern. We agree with you and have revised the conclusion section in order to add the implications for policymakers, practitioners and researchers.

  1. Conclusion section needs to be revised in terms of the after-effects of the study.

Thank you very much for this comment. We realize the conclusion section was no well-developed. We have revised the after-effects of the study.

Reviewer 2 Report

This paper explores the interconnection between the concepts of digital economy and sustainability by using a systematic literature review with a bibliometric analysis (SRLBA). This SRLBA provides insights into opportunities and challenges of the digital economy in a sustainability context to support business professionals’ efforts to integrate advanced technologies in a sustainable development context. Overall, this research topic provides reference opinions for the sustainable development strategy. However, I do have a few minor concerns about the improvement/clarity of the paper.

1. Keywords. Please capitalize the first letter of the keywords. In addition, the number of keywords is small. For example, this article’s systematic literature review with a bibliometric analysis method can be used as a keyword.

2. Introduction. The first paragraph of the introduction appears in the literature format Pan et al twice but should be marked [1]. Check the full text and revise it. In addition, this section should simply summarize the current research status related to the digital economy and sustainability. In addition, the authors should update the latest literatures in the literature review sections, such as doi: 10.1111/itor.13186 and doi: 10.1016/j.cie.2020.106951.

3. Materials and Methods. The text and numbers in the header and tables should be one font size smaller than the main text.

4. Literature analysis: themes and trends. Tables 1 and 2 have already appeared in the second part of Materials and Methods. Thus, using Tables 1 and 2 is not appropriate in this part. Please modify them. Furthermore, the explanation for Table 1 should include the definition of the H index, rather than placing it after Table 2, which appears to be a confusing structure of the paper. Last, Figures 3-5 should provide detailed explanations to illustrate the meaning of the figure.

5. Theoretical perspectives. This section should be appropriately simplified, for example, the concept of the digital economy in section 4.1 is also mentioned in the introduction. The opportunities for sustainability in the digital economy in section 4.4 can also be appropriately simplified and presented in one paragraph.

6. Conclusions. The content in the first paragraph of the conclusion is almost exactly consistent with the abstract. We suggest that the content of this part should be reorganized and improved. Moreover, it is recommended to summarize and provide the conclusion of this article, as the four paragraphs of the conclusion are logically incoherent.

7. References. The DOI format of references 7 and 8 should be modified to the same format as the previous ones. There is much literature with inconsistent DOI formats. Additionally, reference 20 does not provide DOI. Please check all references and unify the format.

This paper explores the interconnection between the concepts of digital economy and sustainability by using a systematic literature review with a bibliometric analysis (SRLBA). This SRLBA provides insights into opportunities and challenges of the digital economy in a sustainability context to support business professionals’ efforts to integrate advanced technologies in a sustainable development context. Overall, this research topic provides reference opinions for the sustainable development strategy. However, I do have a few minor concerns about the improvement/clarity of the paper.

1. Keywords. Please capitalize the first letter of the keywords. In addition, the number of keywords is small. For example, this article’s systematic literature review with a bibliometric analysis method can be used as a keyword.

2. Introduction. The first paragraph of the introduction appears in the literature format Pan et al twice but should be marked [1]. Check the full text and revise it. In addition, this section should simply summarize the current research status related to the digital economy and sustainability. In addition, the authors should update the latest literatures in the literature review sections, such as doi: 10.1111/itor.13186 and doi: 10.1016/j.cie.2020.106951.

3. Materials and Methods. The text and numbers in the header and tables should be one font size smaller than the main text.

4. Literature analysis: themes and trends. Tables 1 and 2 have already appeared in the second part of Materials and Methods. Thus, using Tables 1 and 2 is not appropriate in this part. Please modify them. Furthermore, the explanation for Table 1 should include the definition of the H index, rather than placing it after Table 2, which appears to be a confusing structure of the paper. Last, Figures 3-5 should provide detailed explanations to illustrate the meaning of the figure.

5. Theoretical perspectives. This section should be appropriately simplified, for example, the concept of the digital economy in section 4.1 is also mentioned in the introduction. The opportunities for sustainability in the digital economy in section 4.4 can also be appropriately simplified and presented in one paragraph.

6. Conclusions. The content in the first paragraph of the conclusion is almost exactly consistent with the abstract. We suggest that the content of this part should be reorganized and improved. Moreover, it is recommended to summarize and provide the conclusion of this article, as the four paragraphs of the conclusion are logically incoherent.

7. References. The DOI format of references 7 and 8 should be modified to the same format as the previous ones. There is much literature with inconsistent DOI formats. Additionally, reference 20 does not provide DOI. Please check all references and unify the format.

Author Response

Dear,

Reviewer

This paper explores the interconnection between the concepts of digital economy and sustainability by using a systematic literature review with a bibliometric analysis (SRLBA).This SRLBA provides insights into opportunities and challenges of the digital economy in a sustainability context to support business professionals’ efforts to integrate advanced technologies in a sustainable development context. Overall, this research topic provides reference opinions for the sustainable development strategy. However, I do have a few minor concerns about the improvement/clarity of the paper.

  1. Keywords. Please capitalize the first letter of the keywords. In addition, the number of keywords is small. For example, this article’s systematic literature review with a bibliometric analysis method can be used as a keyword.

We appreciated the reviewer's suggestion and made the changes.

  1. Introduction. The first paragraph of the introduction appears in the literature format Pan et al twice but should be marked [1]. Check the full text and revise it. In addition, this section should simply summarize the current research status related to the digital economy and sustainability. In addition, the authors should update the latest literatures in the literature review sections, such as doi: 10.1111/itor.13186 and doi:10.1016/j.cie.2020.106951.

Thank you for your feedback on the introduction section of the paper. We appreciate your perspective and agree with your suggestions for improvement.

  1. Materials and Methods. The text and numbers in the header and tables should be one font size smaller than the main text.

We appreciate the reviewer's comment and attention to detail. Thus, we proceed to change the tables.

  1. Literature analysis: themes and trends. Tables 1 and 2 have already appeared in the second part of Materials and Methods. Thus, using Tables 1 and 2 is not appropriate in this part. Please modify them. Furthermore, the explanation for Table 1 should include the definition of the H index, rather than placing it after Table 2, which appears to be a confusing structure of the paper. Last, Figures 3-5 should provide detailed explanations to illustrate the meaning of the figure.

Thank you for your valuable feedback regarding this section of our paper. We appreciate your insights and agree that a deep and comprehensive explanation of the figures is crucial.

  1. Theoretical perspectives. This section should be appropriately simplified, for example, the concept of the digital economy in section 4.1 is also mentioned in the introduction. The opportunities for sustainability in the digital economy in section4.4 can also be appropriately simplified and presented in one paragraph.

Thank you for your valuable feedback regarding the theoretical debate in our study. We appreciate your insights, and we have carefully considered your comments. As a result, we have made significant revisions to the Theoretical perspectives part of the Manuscript.

  1. Conclusions. The content in the first paragraph of the conclusion is almost exactly consistent with the abstract. We suggest that the content of this part should be reorganized and improved. Moreover, it is recommended to summarize and provide the conclusion of this article, as the four paragraphs of the conclusion are logically incoherent.

Thank you very much for this comment. We realize the conclusion section was no well-developed. We have revised the all section in order to call your concerns.

  1. References. The DOI format of references 7 and 8 should be modified to the same format as the previous ones. There is much literature with inconsistent DOI formats. Additionally, reference 20 does not provide DOI. Please check all references and unify the format.

Thank you very much for this observation. Indeed, there are several inconsistencies that we resolve immediately. Bibliographical references that do not have a DOI are because it is not yet available.

Reviewer 3 Report

Following changes should be done to improve the quality of this paper further:

* I found abstract is not written well, need to improve to clearly demonstrate the major objective.

* Study examined a sample of 92 studies for the analysis, why not more studies?

* Introduction is short need to expand, and at the end create a new paragraph to show that whats new in this study and how it will make contribution in the existing literature.

* In the methodology part, the quality of figures is not good need to make it colorful.

* Scimago journal & country rank impact factor Table should be 3, need to check all tables numbering sequence wise.

* Why authors used SCOPUS list, why not JCR (Web of Science), Because i found some journals presented in the table are de-listed from the Web of Science. 

* Need to improve the discussion part.

* Precise the conclusion, and also add study limitations.

The moderate editing of English language is required to improve quality further.

Author Response

Dear,

Reviewer

Following changes should be done to improve the quality of this paper further:

  1. I found abstract is not written well, need to improve to clearly demonstrate the major objective.

Thank you for your feedback on the abstract of the paper. We appreciate your perspective and agree its improvement would be beneficial.

  1. Study examined a sample of 92 studies for the analysis, why not more studies?

We appreciate the reviewer's comment. The 92 documents analyzed result from research carried out in the Scopus database, respecting the defined inclusion criteria. This was thus a result generated by the Scopus database.

  1. Introduction is short need to expand, and at the end create a new paragraph to show that whats new in this study and how it will make contribution in the existing literature.

Thank you for your feedback on the introduction section of the paper. We appreciate your perspective and agree that providing additional details on the topic would be beneficial. Expanding the introduction can help set the context, establish the relevance of the study, and provide a more comprehensive understanding for readers. We incorporate more relevant background information to enhance the overall clarity and engagement of the paper.

  1. In the methodology part, the quality of figures is not good need to make it colorful.

We appreciated the reviewer's attention to detail. The images we use in the methodology are those automatically generated by the Vosviewer bibliometric analysis software, so it is not possible to change them.

  1. Scimago journal & country rank impact factor Table should be 3, need to check all tables numbering sequence wise.

We appreciated the reviewer's attention to detail. We corrected the table number to 3.

  1. Why authors used SCOPUS list, why not JCR (Web ofScience), Because i found some journals presented in the table are de-listed from the Web of Science.

We appreciate the reviewer's comment and the opportunity to explain. In addition to the justification given in the article, we understand that this comprehensive database ensures relevant and reliable research with enriched data associated with the academic literature.

Round 2

Reviewer 1 Report

The authors have addressed all my previous comments article is ready for publication

fine

Reviewer 2 Report

The paper has been revised according to the suggestions, and it is recommended to accept it.

Reviewer 3 Report

The quality has been improved.

Need to correct minor grammatical issues.